# Women's experiences of communication with medical staff before and after emergency caesarean birth in Zambia: A qualitative study

Effie Moreblessing Mphande [1,2]*, Doreen Chilolo Sitali[1], Jakub Gajewski[3,4]

1 Department of Health Promotion and Education, School of Public Health, University of Zambia, Lusaka, Zambia, 2 Department of Social Work and Community Development, University of Johannesburg, South Africa, 3 Institute of Global Surgery, School of Population Health, Royal College of Surgeons in Ireland, 4 Centre for Global Surgery, Stellenbosch University, South Africa

☉ DCS and J.G. are joint senior authors

* effiemphande@gmail.com, emphande@uj.ac.za

## Abstract

### Background

Communication between patients and staff during emergency caesarean birth is important for ensuring positive outcomes and reducing the negative psychological impact of the procedure. Communication failures have been linked to obstetric violence, mistrust, and post-traumatic stress disorder. This study aimed to explore patient-healthcare provider communication before and after emergency caesarean birth at the University Teaching Hospitals, Women and Newborn Hospital in Lusaka, Zambia.

### Methods

This study employed a qualitative phenomenological design to explore the lived experiences of women who had undergone emergency caesarean birth. Interviews were conducted with 30 women who were purposively sampled from the hospital's wards. An inductive thematic analysis, which involved transcribing interviews, reading and rereading transcripts, coding, categorising similar codes and developing themes, was used for data analysis.

### Results

Thematic analysis yielded four primary themes: mode of communication, emergency caesarean birth communication experience, consequences of inadequate communication and information, and barriers to effective communication. Communication between healthcare providers and women who underwent emergency caesarean birth was inadequate, untimely and lacked detail about the surgical procedure. As a result, women felt afraid, angry, and anxious, resulting in a

**Data availability statement:** Anonymized transcripts have been uploaded as Supporting Information. All additional relevant data are available from University of Zambia Biomedical Research Ethics Committee (unzarec@unza.zm).

**Funding:** The author(s) received no specific funding for this work.

**Competing interests:** The authors have declared that no competing interests exist.

sense of worthlessness and helplessness. The use of medical jargon by healthcare providers, misconceptions about caesarean birth, the presence of pain and poor staff attitudes towards mothers were identified as some of the communication barriers.

## Conclusion

The findings highlight systemic gaps in provider-patient communication during emergency caesarean birth, influenced by workload pressures, staff shortages and power dynamics. Interventions are needed to promote respectful maternity care through training in patient centred communication, use of simple language, and addressing structural barriers at the University Teaching Hospitals Women and Newborn Hospital. Clear communication can help to improve the overall experience of caesarean birth.

## Introduction

Caesarean birth remains the most appropriate option for several obstetric problems [1] and contributes greatly to safe motherhood. Despite an increase in caesarean birth, there is a paucity of literature documenting women's experiences of communication with medical staff before and after emergency caesarean birth. As a result, communication between women undergoing emergency caesarean birth and medical staff (Nurses/Midwives and obstetricians) is crucial for ensuring positive outcomes and reducing the negative psychological impact of the procedure [2]. Communication failures during obstetric care have been identified as contributing factors to obstetric violence, erosion of trust in healthcare systems, and the development of post-traumatic stress disorder (PTSD) among women [3–6]. These issues are particularly pronounced in low and middle income countries where healthcare systems face significant resource constraints. Effective communication can help to alleviate women's fears, reduce anxiety and improve the overall experience of caesarean birth [3–7]. Clear, timely and empathetic communication enables women to understand the reasons for the procedure, the risks and benefits, and the steps taken during the procedure [8]. This can lead to better patient compliance, satisfaction and trust in the medical staff [7,9]. Furthermore, good communication can help women feel more in control and empowered, which can help them cope better with the procedure [8]. Poor communication can lead to negative experiences, dissatisfaction with care and mistrust towards the medical staff [6–10].

This study aimed to explore the experiences of women concerning communication with medical personnel before and after an emergency caesarean birth at the University Teaching Hospital in Lusaka, Zambia. This research aims to contribute to the understanding of the current communication practices and gaps in maternal health identifying gaps and informing strategies to improve the patient experience for women who undergo emergency caesarean birth in similar settings.

## Materials and methods

### Study design and sampling

We employed a phenomenological study design to understand women's experiences of communication with medical personnel before and after emergency caesarean birth at the University Teaching Hospitals Women and Newborn Hospital (UTHWNH). Purposive sampling was used to select 30 women who had particular characteristics from the hospital's five wards. The sample size of 30 participants was determined based on established qualitative research guidelines suggesting that phenomenological studies typically involve 20–25 participants to achieve thematic saturation [11]. This slightly larger sample was used to ensure a diverse range of perspectives and achieve thematic saturation. During data collection, thematic saturation was assessed iteratively; data collection ceased when no new themes, codes, or patterns emerged from subsequent interviews, which occurred after interviewing 28 participants. Two additional interviews were conducted to confirm saturation.

### Participant recruitment

The participants were approached individually after formal introductions to the study done in the wards. Participants included women who had undergone emergency caesarean birth, admitted to the hospital, 48 hours postoperatively, conscious and signed the consent form before being taken to theatre. Women who were critically ill after the emergency caesarean birth, 0–1 day post-operative and those who had stillbirths were excluded. Information regarding demographical characteristics was obtained from the patients' files. All participants received care within the public healthcare system at UTHWNH; private healthcare patients were not included in this study. Fig 1 shows the number of participants approached, excluded, and included in the study.

### Data collection

An interview guide with open-ended questions was used to explore women's experiences of communication with healthcare providers before and after emergency caesarean birth (see S1 File). Face-to-face, in-depth interviews were conducted by the lead author between 17 March and 11 April 2021, within 48 hours post-operatively. To ensure participant privacy, interviews were conducted in private treatment rooms or private rooms within the hospital wards.

The interview guide was professionally translated into the two main languages spoken in the capital city by a translator from the Ministry of Education. Interviews were conducted in English, Bemba or Nyanja depending on participant preference, with one held in Tonga, and each session lasted approximately 30–45 minutes. Narratives were audio-recorded using a digital voice recorder. Data collection and analysis were conducted concurrently. Data collection ceased once no new information emerged, indicating thematic saturation. Participants were asked to describe their experiences of communication with healthcare providers both before and after undergoing emergency caesarean birth. A triangulation approach was employed to enhance the credibility of the findings. After patient data collection, consultations were held with 17 medical professionals. These included obstetricians and gynaecologists with clinical and managerial responsibilities, registrars and mid wives (see S2 File). The discussions were used to counter-check and contextualise the clinical experiences reported by patients as well as understand the hospital operations. While consultations were held with healthcare providers to validate operational flow and counter-check patient reported data, these discussions were solely used for contextual validation and triangulation. Consequently, the findings of this study are reported exclusively from the patient perspective, as staff input was intended to ensure methodological accuracy rather than serve as a primary outcome. To ensure trustworthiness, data was also cross-checked with the participants after the interviews to verify their responses. We followed the COREQ checklist for reporting qualitative data in this study [12] and adhered to the Plos' questionnaire on inclusivity in global research. The checklists are provided as supporting information (see S3 File and S4 File).

Potential participants approached (n = 35)
participants approached (n = 35)

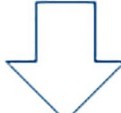

Excluded (n = 5)
•Not eligible (n = 3)
•Declined to participate (n = 2)

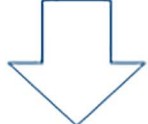

Consented (n = 30)
Completed interviews (n =30)
Included in final analysis (n = 30)

**Fig 1. Flowchart of participant recruitment.**

### Ethical approval and consent to participate

Ethical approval was obtained from the University of Zambia Biomedical Research Ethics Committee (REF. No. 1445–2020) and National Health Research Authority (Ref No: NHRA00011/4/03/2021). Permission was also sought from the Lusaka Provincial Health Office (PHOLSK/101/8/11). Administrative permission to conduct the study at the University Teaching Hospitals, Women and Newborn Hospital was granted by the Senior Medical Superintendent and Unit Head, Clinical Research and Quality Improvement. Specific permission to access the Operating Theatre Registers was obtained from the Theatre Superintendent, while access to individual patient medical files was authorised by Nurse Managers

(Sisters-in-charge) in charge of the respective wards at the University Teaching Hospitals, Women and Newborn Hospital. Written informed consent was obtained from all participants before they took part in this study. As part of the informed consent process, the purpose of the study was explained to participants. Before interviews, they were given an opportunity to ask questions and sign the consent form. Participants were informed that their participation was voluntary and that they could withdraw from the study at any point if they wished to. They were assured that their withdrawal from the study would not affect access to health services.

### Data analysis

The recordings were transcribed verbatim and field notes were typed and saved as a Word document. Interviews, which were done in local languages were transcribed and translated into English by personnel from the Zambia National Broadcasting Corporation (ZNBC) home language section. Data collection and analysis occurred simultaneously. Data were analysed using a thematic analysis approach [11]. An inductive iterative process of reading and rereading the transcripts produced subcategories for data analysis within research areas of interest. This was followed by developing preliminary codes. 55 initial codes were developed. Similar codes were grouped and similar patterns were put together resulting in 19 final codes. After that, themes were created and a coding tree was developed. NVIVO 11 qualitative data analysis software was used to facilitate the analysis [13]. Direct quotes were used to illustrate specific concepts. Anonymised transcripts and codebooks used for analysis are provided as supporting information (S5–S8 Files).

### Research team and reflexivity

The interviews were conducted by the lead author (EMM), a female researcher who at the time of the study was a Master of Public Health (MPH) student at the University of Zambia. Before the study, the lead author had completed coursework for the Master's degree in Public Health, Health Promotion and Education. The researcher possesses a diverse multi-disciplinary background, holding a Master's degree in Peace, Leadership and Conflict Resolution, a Bachelor of Arts in Communication Science, and a Diploma in Journalism, Public Relations and Advertising. Furthermore, she holds a certificate in Qualitative Research Methods for Global Public Health. The selection of the research topic was influenced by the researcher's communication professional background. The journalism background helped balance empathy and objectivity as well as upholding ethical standards. While conducting the study, EMM reflected on her positionality, supported by conversations with the other researchers. This was important to minimise bias.

The use of in-depth interviews enabled to probe further and ask follow-up questions. In-depth interviews also allowed the researchers to capture the participants' perspectives in detail. Prior to the interviews, none of the participants were known to EMM, either in a personal or professional capacity. To enhance credibility, the study was supported by JG and DCS, both with vast experience and expertise in qualitative research in Zambia. The transcripts and audio recordings were shared with them for their analytical input and to agree on codes and themes between the research team.

## Results

### Participants' attributes

Table 1 summarises their demographic characteristics. The majority of participants were aged between 30 and 39, married and school leavers. 19 of them had an emergency caesarean birth for the first time while eight had the operation for the second time and three were being operated on for the third time.

The thematic analysis revealed four main themes and their associated sub-themes (see Table 2). These themes and subthemes are presented systematically with supporting quotations from participants. Fig 2 shows the organisation of themes and subthemes. The study presents study title on the left, main themes in the middle, and subthemes on the right, illustrating the organisation of the findings.

 

**Table 1. Participant characteristics.**

| Characteristic | N | % |
|---|---|---|
| **Age (years)** | | |
| <25 | 1 | 3.3 |
| 25-29 | 8 | 26.7 |
| 30-39 | 18 | 60.0 |
| ≥40 | 3 | 10.0 |
| **First-time mother** | | |
| Yes | 9 | 30.0 |
| No | 21 | 70.0 |
| **Number of emergency caesarean births** | | |
| 1 | 19 | 63.3 |
| 2 | 8 | 26.7 |
| 3 | 3 | 10.0 |
| **Level of education** | | |
| No formal education | 7 | 23.3 |
| Primary education | 7 | 23.3 |
| Secondary education | 5 | 16.7 |
| Certificate/Diploma | 6 | 20.0 |
| Degree/Postgraduate | 5 | 16.7 |

**Table 2. Summary of themes and sub-themes.**

| Theme | Sub-themes |
|---|---|
| 1. Mode of communication | Verbal, nonverbal, and written communication. |
| 2. Emergency caesarean birth communication experience | Inadequate information, timeliness of communication; Lack of detail. |
| 3. Consequences of inadequate communication | Emotions; (anxiety, fear, nervousness, sadness, anger, frustration, feelings of helplessness, a sense of worthlessness, and surprise). Development of coping mechanisms; Information-seeking behaviours |
| 4. Barriers to effective communication | Medical jargon; Language barriers; Misconceptions about caesarean birth; Pain; Staff attitude |

## Mode of communication

This theme delineates the diverse channels through which healthcare providers communicated with women who underwent emergency caesarean birth. While communication was inherently multi-modal – encompassing verbal, nonverbal, and written channels – face-to-face interaction emerged as the predominant approach.

**Verbal communication.** Verbal communication was the primary mode through which women received information from healthcare providers. It involved spoken words. Communication was often restricted to information provision, convincing the women to accept emergency caesarean birth and avoid complications. Communication between healthcare providers and women took place in the admission rooms of the labour ward, high dependency unit or obstetrics intensive care unit before they were taken to the theatre.

**Non-verbal communication.** Nonverbal communication, which involved facial expressions, tone of voice, voice volume, voice pace, eye contact, silence, distance (physical space), body movements and posture, hand gestures and

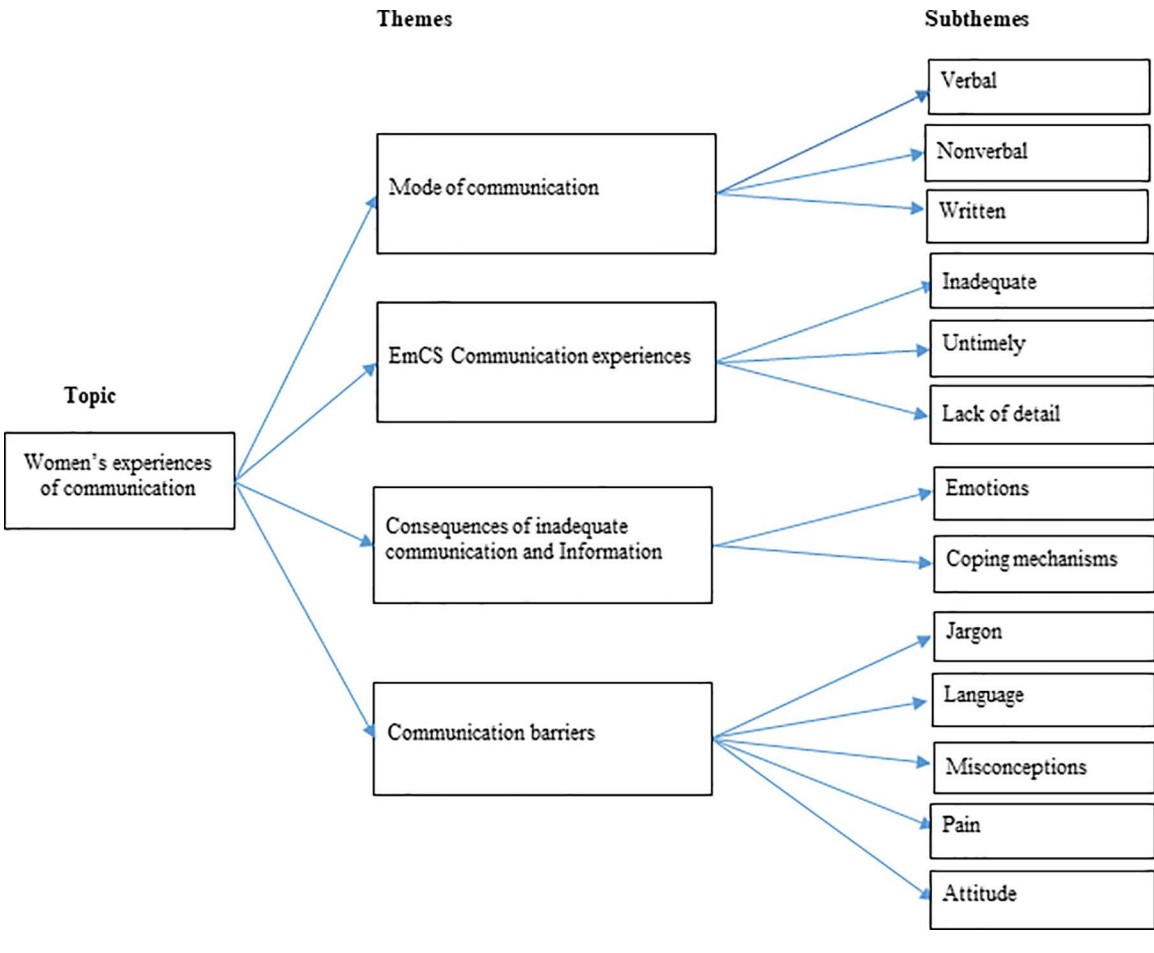

**Fig 2. Coding tree.**

touch, played a significant role in women's experiences of communication with healthcare providers before and after emergency caesarean birth. Women reported that non-verbal cues from healthcare providers could either reassure or intimidate them.

**Written communication.** Written communication was in the form of obtaining written consent from the women before they could be operated on. In most cases, the consent form was signed in the absence of the medical doctor. Communication between medical personnel and women who underwent emergency caesarean birth was usually firstly verbal and thereafter written consent was obtained by midwives. In the words of one participant:

*"The doctor told me I was going to have caesarean section then the nurses who were preparing me for theatre are the ones who gave me the consent form," (Mother 15).*

The women provided written consent by appending their signatures or putting a thumbprint on the form. The consent form was in English. According to study participants, the purpose of the consenting process was to explain the indication for emergency caesarean birth, the risks of not performing the operation and obtaining verbal and written consent.

### Emergency caesarean birth communication experience

This theme describes women's perceptions of the quality and adequacy of communication they received. Communication between healthcare providers and women who underwent emergency caesarean birth was reported to be inadequate, untimely, and lacking detail about the surgical procedure.

**Inadequate communication.** Inadequacy emerged as a key challenge influencing women's experience of communication with healthcare providers before and after emergency caesarean birth. The majority of the study participants said communication was incomplete in the sense that it lacked an explanation of the women and baby's condition, classification of emergency caesarean birth, amount of pain women would experience after the procedure, anaesthesia side effects after the operation, consequences of not adhering to post caesarean birth advice and recovery trajectory. For example, one mother indicated that she had not been advised to lay down for eight hours after the emergency caesarean birth. She reported that she only learnt about this after she questioned medical personnel on the cause of her excruciating headache.

*"You can imagine the damage I would have done to myself. Just that piece of information, which they know and could have communicated," (Mother 1).*

Most of the study participants said communication also lacked clarity. Regardless of the emergency, women expected medical personnel to communicate clearly before and after the surgical procedure.

**Timeliness of communication.** Some study participants reported that communication before and after emergency caesarean birth lacked timely and regular updates. The women described long periods without being informed about when they would be taken into theatre. For instance, mothers who had been notified of the need for emergency caesarean birth but were not taken into theatre immediately desired to be communicated to about the reason for the delay. Some women said they expected healthcare providers to communicate the delay immediately after they noticed that the operation was not going to take place at once. They thought they needed to be taken into theatre immediately after being told that they were required to give birth through emergency caesarean birth. Not being taken to the theatre instantly left them wondering whether their case was an emergency or not.

Some of the participants wanted to be given a timeframe in which they would be taken to the theatre so that they could determine when to eat. Some women reported fasting for over 17 hours before being taken to the operating room.

**Lack of detail.** Furthermore, some of the interviewed mothers reported that often the communication lacked detail about the medical procedure in that it did not contain the steps to be taken during surgery and the duration of the operation. They indicated that it also lacked information about risks associated with the operation.

*"You have to make somebody understand what they are getting themselves into and you can mention the odds of a successful emergency caesarean section because I didn't even know anything about the procedure, how much risk do I have, is it possible that it might not be successful, I didn't even have that information you know so if I who is a health professional don't know what more somebody for instance these guys from rural areas who have never even gone to school?" (Mother 2).*

However, some women reported that they were told about the need for emergency caesarean birth, the procedures involved and the consequences of not being operated on. One participant pointed out that before emergency caesarean birth she had been informed that she would experience pain for two weeks after the operation.

*"They explained to me what I went through before they conducted a caesarean section on me. I was told what I would experience, I was told the pain would be there for about two weeks," (Mother 10).*

In several instances, rapport was lacking between healthcare providers and the mothers. Most mothers viewed the uncommunicative attitude of medical personnel as a hindrance to asking questions or seeking clarity. Women felt that some of the staff were impolite making it difficult to communicate with them. Some women reported that they did not want to ask questions for fear of upsetting medical staff. This was common among all the participants regardless of their level of education.

*"In my case I was in pain so I didn't want to ask questions for fear of upsetting them because I was the one who was in need,"* (Mother 12).

### Consequences of inadequate communication and information

This theme covers the emotional and behavioural responses that resulted from inadequate communication. Women experienced a range of emotions including anxiety, fear, nervousness, sadness, anger, frustration, feelings of helplessness, a sense of worthlessness, and surprise. This affected effective communication with healthcare providers. Two sub themes namely emotions and coping mechanisms emerged from the data.

**Emotions.** Women described a range of emotions before and after undergoing emergency caesarean birth, including anxiety, fear, nervousness, sadness, anger, frustration, surprise, feelings of helplessness, and a sense of worthlessness. These emotions often reflected their challenges with uncertainty about what to expect after surgery, the distress caused by side effects of anaesthesia, and difficulties arising from inadequate and untimely communication, lack of detail, and lack of clear explanations from healthcare providers.

**Anxiety.** Some women experienced anxiety related to uncertainty about post-emergency caesarean birth outcomes and side effects of anaesthesia.

*"I didn't understand why I was feeling very cold but later on after I asked they told me it was the medication that it would clear afterwards but with me I felt hmm this was not normal am I going to be ok because when I came (to the ward) I told mum to put blankets and close the windows but nothing helped not until after everything. I think if someone had told me what to expect after caesarean section that would have helped,"* (Mother 16).

*"I felt like I was going to die. Being an emergency no one prepares for an emergency but maybe the time the nurse was preparing me to go into theatre maybe she should have given me a hint about one or more things that are important so that my mind is ready for it,"* (mother 15).

**Fear.** Women reported fear in response to emergency caesarean birth, with some describing the experience as terrifying.

*"It is terrifying because you are asking yourself am I going to come back alive, is my baby going to be ok or are we both going to be ok, so it is kind of traumatic and terrifying at the same time. At least maybe if the mind is prepared it would help,"* (Mother 20).

*"I was scared, I felt like this is the end you know mmm ah I was very scared I can't lie I was very scared. I just thank God that I am still alive today. It is just by God's grace. I had lost hope but I thank God that everything worked out for good,"* (Mother 21).

**Frustration.** Frustration was reported when women felt uncertain about their own condition and the wellbeing of their babies before emergency caesarean birth. Many described feeling frustrated, particularly when they experienced delays and lack of explanation from healthcare providers.

**Helplessness.** Helplessness occurred when some women felt they had no control over their care or outcomes.

*"You are just at the mercy of healthcare providers, you are just being experimented on you know, had I been on the upper hand it is not something I would really advocate for," (mother 2).*

**Worthlessness.**  Some women experienced feelings of worthlessness when they perceived that their needs and concerns were undervalued by healthcare providers. This emotion arose from experiences of lack of attention during care. Feeling unimportant contributed to a sense of diminished self-worth and emotional distress.

*"I felt neglected, like my case was not important because I was also fighting for my life whereby they are just quiet, it made things worse,"(Mother 20).*

*"Ah I felt like I was just an object ha ha like I was just an object on a bed, like I had no feelings, yeah like an object like I had no feelings whatsoever because this was my life they were talking about and I wasn't even involved in the process of delivery or being communicated to"(Mother 1).*

**Nervousness.**  Nervousness was reported prior to emergency caesarean birth, linked to uncertainty of the surgical procedure.

**Surprise.**  Surprise emerged as a notable emotional response among women. Some women described surprise arising from sudden changes in their medical condition, and unexpected emergency caesarean birth. Participants reported feeling startled when they were told that they required emergency caesarean birth. Women felt unprepared for the surgical procedure. One participant shared:

*"I just came for a normal check-up and I was told it is important that I go through the C-section if the normal one doesn't go through. So it was not planned that this day I would have an operation," (Mother 10).*

**Coping mechanisms.**  Women employed various coping mechanisms to cope with emotional distress and their communication experiences. The coping strategies enabled them to maintain a sense of stability. These included seeking information from the internet, friends, relatives and those who had emergency caesarean birth as postnatal and antenatal mothers were being nursed in the same ward. This was common among first-time mothers and those who had the surgical procedure for the first time regardless of their level of education.

### Barriers to effective communication

This theme identifies the factors that hindered effective communication between healthcare providers and women who underwent emergency caesarean birth. Multiple barriers were identified, including medical jargon, language differences, and misconceptions about caesarean birth, pain, and staff attitude.

**Use of medical jargon.**  The majority of the women reported that unfamiliar medical terms were used before and after emergency caesarean birth, making it difficult for them to understand what medical personnel communicated. They reported that technical language was used without explanation, leading to anxiety, and reduced engagement in decision making. The majority of women reported that some healthcare professionals did not know how to explain the medical terms they were using in an understandable, lay language manner. They desired medical personnel to explain their condition so that they could relay the information to whoever would be taking care of them. Others wanted a situation where they would be made to understand medical issues so that they could explain to family members and friends the reason they had the surgical procedure.

**Language differences.**  Language was also identified as a barrier to communication. Some of the participants reported difficulties arising from interactions conducted in languages they were not fully proficient in, which limited their comprehension. Some women reported that medical personnel did not use local languages they understood when

communicating with them. They also stated that the hospital staff used local languages popular in the capital city, which other patients from more remote places could not understand. This sentiment was raised by mothers in the low-cost ward regardless of their level of education. Mothers wanted medical personnel to use a diversity of language when giving out information to patients. As one participant explained:

*"For this hospital they use a lot of Bemba and Nyanja and we see a lot of people in here, we are not all Bembas and Nyanjas yes. Maybe if there could be a diversity of language so that medical staff can give information that everybody requires regardless of the language the patients speak," (Mother 28).*

Overall, language barriers constrained meaningful communication and reinforced power imbalances between women and healthcare providers, further contributing to emotional distress.

**Misconception about emergency caesarean birth.** The other barrier to communication was misconceptions women had about emergency caesarean birth. Some perceived emergency caesarean birth as an act of God (uncontrollable natural force), which only God could choose or inflict on mothers. This view was expressed by those who had staunch Christian faith regardless of their level of education. In addition, emergency caesarean birth was seen as an act of kindness especially among those with low levels of education.

*"It is by the grace of God that they chose this path for me, I did not choose it for myself," (Mother 10).*

*"God has the final say. God guided me on what to follow and accept caesarean section," (Mother 18).*

In consequence, some women felt that medical doctors had acted kindly by conducting emergency caesarean birth to save their lives and that of their babies.Learning about the need to have an emergency caesarean birth alarmed some of the women and provoked thoughts of being in a serious condition. Before being operated upon, all the women knew about caesarean birth, however, the majority did not know that there was an emergency caesarean birth. The mothers were under the impression that all those who gave birth through caesarean birth did so on their own accord.

**Pain.** The presence of pain was also identified as a barrier to communication. Mothers who were in advanced labour desired to be taken for a caesarean birth before being provided with information as the pain was unbearable. They indicated that they were tired and just wanted the pain to go away. This was common among mothers with low levels of education and who had been referred from lower-level hospitals. One shared:

*"Hmm…..I was in pain, I didn't know anything when I went in theatre but I just signed the form because I really needed help," (Mother 14).*

**Staff attitude.** Staff attitude towards mothers also hampered communication. There was a reported feeling of lack of respect from healthcare providers. The majority of the participants felt that they were spoken to rudely and intimated that it would be impossible to have proper communication when the other party was disrespectful. A sense of mistrust was perpetuated by a lack of respect from healthcare providers in the short time the women were at the hospital. Some women indicated that they were shouted at or yelled at by some nurses instead of being explained to.

*"There were some things that I did not understand but I was surprised that I was shouted at. Some of us don't understand medical issues because it is not in our line of work. So it is better they explain instead of shouting. Like in my case, this was my first emergency caesarean section but they shouted at me on things that I didn't even know or understand," (Mother 8).*

Other participants expressed similar concerns mentioning explicitly some of the language used:

*"I would rather they explain that when you do this and that it will affect you this way than some other nurses come out to say <I am not the one who impregnated you>., Such words are not good. So, I would really love to see a change on that,"* (Mother 23).

However, not all study participants reported similar concerns, five women intimated they were treated in a dignified manner. As one woman explained:

*"They treated me well and spoke to me nicely as a patient is supposed to be handled,"* (Mother 22).

## Discussion

This study explored women's experiences of communication with healthcare providers before and after emergency caesarean birth and identified several themes including mode of communication, emergency caesarean birth communication experience, consequences of inadequate communication, and barriers to effective communication. This is the first study in Zambia investigating communication during emergency caesarean birth in a tertiary hospital. It offers valuable insights to inform strategies aimed at improving the quality of emergency caesarean birth care, with potential applicability across the country and beyond. Furthermore, this study provides new insights into how patients experience communication in emergency caesarean birth. These findings extend existing literature by foregoing patients' voices and illustrating how communication practices shape psychological experiences surrounding emergency caesarean birth. While rooted in the Zambian context, particularly at the University Teaching Hospitals, Women and Newborn Hospital in Lusaka, the findings align with existing literature from other sub-Saharan African countries, suggesting broader relevance across similar healthcare systems. Consistent with findings from a study in Malawi [14], face-to-face communication—both verbal and nonverbal—emerged as a key mode of interaction between women and healthcare providers. Such communication allows patients to interpret nonverbal cues [15] and anticipate providers' responses. In our study, several women reported refraining from asking questions due to perceived nonverbal signals suggesting that doing so might upset healthcare staff. This reflects entrenched power imbalances that discourage open dialogue—a trend similarly reported in Uganda, Tanzania, and Nigeria [16–19]. These studies demonstrate how patients rely on nonverbal indicators such as gestures, facial expressions, posture, and attire to gauge whether the clinical environment is welcoming or intimidating [20].

The findings suggest that poor communication during emergency caesarean birth is not merely an individual provider issue but reflects broader systematic challenges within the healthcare system. Several factors may contribute to the communication gaps observed. First, healthcare worker shortages and workload in tertiary facilities like UTHWNH may limit the time available for meaningful patient-provider interactions. When healthcare providers are managing multiple emergencies simultaneously, comprehensive communication with each patient may become secondary to clinical priorities. Second, power dynamics between healthcare providers and patients, particularly in hierarchical healthcare cultures, may inhibit open dialogue and patient questioning. Women's reluctance to ask questions for fear of upsetting medical staff reflects an environment where patients feel subordinate rather than empowered participants in their care. Third, inadequate training in patient centred-communication skills during medical and nursing education may leave providers ill equipped to effectively communicate during high-stress situations such as obstetric emergencies.The findings have significant implications for respectful maternity care frameworks, which emphasise women's rights to dignified, respectful, and equitable treatment during childbirth [21,22]. The World Health Organization's framework on respectful maternity care identifies effective communication as a core component of quality maternal healthcare. Our findings reveal that communication practices at UTHWNH fall short of these standards in several ways. The use of disrespectful language (e.g., "I am not the one who impregnated you"), shouting at patients, and failure to provide adequate information all represent violations of respectful care principles. Addressing these issues requires a multipronged approach that includes institutional policy changes, provider training, and mechanisms for accountability.

At UTHWNH, women's hesitation to speak up suggests that hierarchical and non-inclusive communication practices may inhibit patient engagement in maternal care. Most participants expressed dissatisfaction with the quality of communication from healthcare providers, often linking negative emotions to poor interactions. Similar to findings from other regional studies [10,19], many women attributed their distressing birth experiences to a lack of information and ineffective communication. One possible explanation is that healthcare providers may only communicate what they perceive as critical—overlooking the importance of explaining the reasons for surgery, operative risks, recovery expectations, and potential complications.

Women in our study explicitly expressed the need to understand the duration of the procedure, outcomes for their baby, and the consequences of non-adherence to post-operative advice, echoing findings from other studies in sub-Saharan Africa [23,24]. The absence of clear communication before and after the emergency caesarean birth contributed to widespread anxiety and mistrust. For example, some women questioned whether their surgery was truly urgent due to unexplained delays. Unaware that other high-risk cases were being triaged, they interpreted the lack of information as negligence. This mirrors findings from a study in the US, where women also reported frustration with inadequate communication throughout obstetric emergencies [25].

Furthermore, the emotional state of patients, marked by fear, anxiety, and a lack of emotional support, likely influenced both the reception and perception of communication. Similar associations between emotional distress and impaired communication have been documented in Spain [26], where patients' apprehension and emotional instability were shown to disrupt effective interactions with healthcare staff.

In response, many women developed coping strategies. They sought information from peers in the maternity ward, as well as online sources, friends, and relatives. This pattern aligns with previous research showing that in the absence of adequate counselling from healthcare providers, women turn to informal networks and digital resources [10,27]. Other studies suggest that patients increasingly consult the Internet as their first source of health information, often before interacting with medical personnel [27].

Our study identified several barriers to effective communication between providers and women undergoing emergency caesarean birth. These included the use of medical jargon, language differences, and misconceptions about caesarean birth, pain, and staff attitudes. Many participants struggled to understand healthcare messages due to complex terminology, a challenge also reported in other studies across the region. Language barriers further complicate communication, especially in multilingual settings such as Zambia, where several local languages are spoken. Previous studies and research from other countries [28–30] confirm that language discordance remains a persistent obstacle in healthcare delivery. Pain was another significant barrier. In some cases, severe discomfort prevented women from engaging meaningfully with providers or processing the information given. Moreover, healthcare staff were sometimes reluctant to provide explanations during periods of intense pain, deeming such communication ineffective, a view supported by similar findings in Malawi [14]. While this rationale may be understandable, it highlights the need for timing and sensitivity in patient interactions.

Lastly, the attitude of some healthcare providers emerged as a key factor affecting communication. Reports of disrespectful or dismissive behaviour suggest a broader cultural issue within the facility, contributing to breakdowns in provider-patient interaction. Similar issues have been reported in other studies from Zambia and neighbouring countries [8,31], underscoring the importance of addressing healthcare culture and provider behaviour as part of efforts to improve maternal care.

### Strengths and limitations

The strength of this study is that it is the first to explore women's experiences of communication with healthcare providers before and after emergency caesarean birth in Zambia. This study provides novel insights into maternal care as there is limited research in this area. The interviews were conducted by a non-clinician researcher, which may have helped

participants feel more comfortable sharing their experiences openly without concerns about clinical judgement or consequences. The use of member checking enhanced the trustworthiness of the findings. However, there are several limitations in our study. A potential limitation of this study is social desirability bias associated with the hospital setting. Participants may have felt inclined to provide responses that they perceived as more favourable to the hospital. Furthermore, the study only documented the views of patients and did not represent the views of the healthcare providers which could have shed a different light on the issue of communication in the situation of emergency. However, this is not a weakness since the researchers' scope was limited to women, yet further work on the voice of the providers regarding medical communication is needed to fully understand this phenomenon. Another limitation is that for interviews done in the local languages some medical terms for example *mwadzidzidzi* (emergency) could not be understood by some women. To mitigate this, short descriptive terms (colloquial language) such as "fast fast" were used by the lead researcher. However, there is no guarantee that patients fully understood the meaning. Lastly, it is a single-site study which limits its generalisability, and more research is needed to understand whether similar experiences are common in other tertiary facilities and at other levels of care. To facilitate the translation of these findings into clinical practice, a summary report will be submitted to the University Teaching Hospitals Women and Newborn Hospital Senior Medical Superintendent and Head of Department of Obstetrics and Gynaecology.

## Conclusion

This study highlights the critical role of communication in shaping women's experiences of emergency caesarean birth, demonstrating how interpersonal dynamics, emotional well-being, and systemic barriers influence patient engagement and satisfaction. The findings reveal significant gaps in provider-patient communication at the University Teaching Hospitals, with practical implications for improving maternal care. Based on these findings, we recommend that healthcare institutions implement training programmes focused on patient-centred communication skills for all staff involved in maternal care. Healthcare providers should be encouraged to use simple, jargon free language and ensure information is available in local languages. Institutional policies should promote respectful maternity care principles and establish mechanisms for patient feedback. The findings resonate with evidence from other sub-Saharan African countries, pointing to a regional challenge that warrants broader investigation. Future research should explore healthcare provider perspectives, evaluate interventions to improve provider-patient communication, particularly in high-stress, multilingual, and resource-limited settings. Comparative studies across diverse healthcare systems in sub-Saharan Africa could further illuminate context-specific and shared barriers, while implementation research is needed to evaluate strategies that promote respectful, inclusive, and patient-centred communication in maternal care.

## Supporting information

**S1 File. Interview guide for women.**
(PDF)

**S2 File. Interview guide for healthcare providers.**
(PDF)

**S3 File. Inclusivity in global research questionnaire checklist.**
(PDF)

**S4 File. COREQ checklist.**
(PDF)

**S5 File. Women anonymised interview transcripts.**
(PDF)

**S6 File. Codebook for interviews with women.**
(DOCX)

**S7 File. Healthcare providers anonymised interview transcripts.**
(PDF)

**S8 File. Codebook for discussions with healthcare providers.**
(PDF)

## Acknowledgments

The research team would like to thank the participants who in the most sensitive period in their lives agreed to participate in this study.

## Author contributions

**Conceptualization:** Effie Moreblessing Mphande, Doreen Chilolo Sitali, Jakub Gajewski.

**Data curation:** Effie Moreblessing Mphande.

**Formal analysis:** Effie Moreblessing Mphande, Jakub Gajewski.

**Investigation:** Effie Moreblessing Mphande.

**Methodology:** Effie Moreblessing Mphande, Doreen Chilolo Sitali, Jakub Gajewski.

**Supervision:** Doreen Chilolo Sitali, Jakub Gajewski.

**Validation:** Doreen Chilolo Sitali, Jakub Gajewski.

**Writing – original draft:** Effie Moreblessing Mphande.

**Writing – review & editing:** Doreen Chilolo Sitali, Jakub Gajewski.

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
