## [Decision Letter · Decision Letter 0]

25 Nov 2025

Dear Dr. Mphande,

Thank you for submitting your manuscript to PLOS ONE. After careful consideration, we feel that it has merit but does not fully meet PLOS ONE’s publication criteria as it currently stands. Therefore, we invite you to submit a revised version of the manuscript that addresses the points raised during the review process.

We look forward to receiving your revised manuscript.

Kind regards,

Sunita Panda, PhD

Academic Editor

PLOS ONE

**Journal Requirements:**

3. In the online submission form you indicate that your data is not available for proprietary reasons and have provided a contact point for accessing this data. Please note that your current contact point is a co-author on this manuscript. According to our Data Policy, the contact point must not be an author on the manuscript and must be an institutional contact, ideally not an individual. Please revise your data statement to a non-author institutional point of contact, such as a data access or ethics committee, and send this to us via return email. Please also include contact information for the third party organization, and please include the full citation of where the data can be found.

5. Please ensure that you refer to Figures 1 and 2 in your text as, if accepted, production will need this reference to link the reader to the figure.

**Additional Editor Comments**

Thank you for submitting your manuscript. Both reviewers have provided detailed feedback and have suggested revisions. Therefore, I invite you to respond to the reviewers' comments and submit your revised manuscript.

Reviewers' comments:

Reviewer's Responses to Questions

**Comments to the Author**

1. Is the manuscript technically sound, and do the data support the conclusions?

Reviewer #1: Yes

Reviewer #2: Partly

2. Has the statistical analysis been performed appropriately and rigorously?

Reviewer #1: N/A

Reviewer #2: No

3. Have the authors made all data underlying the findings in their manuscript fully available?

Reviewer #1: Yes

Reviewer #2: Yes

4. Is the manuscript presented in an intelligible fashion and written in standard English?

Reviewer #1: Yes

Reviewer #2: Yes

Reviewer #1: The authors are to be commended for this study which addresses the experiences of women experiencing emergency caesarean birth.

Well done on using terminology for caesarian birth but there is inconsistency in the text where caesarean delivery is used in some situations. Best to be consistent or else explain why different terms are used.

Table 1 could be presented in a format that is easier to read i.e. number of first time mother, age groups, education level etc.

Did all participants have the same level of care, just wondered if some had private health care and if that was reflected in the interviews?

L 244 Fasting a better term to use than starving

Reviewer #2: Dear Author

consider the following ponts for improvement.

Abstract:

- can add important details, eg. design- phenomenological qualitative design, inductive thematic analysis...etc

- Specify key themes in results can added...

Introduction:

- Add about key global issues e.g.,

- Communication failures contribute to obstetric violence, mistrust, and PTSD...etc

- need of the study or knowledge gap to be added to justify the need of the study.

Material and methods:

- add: justification- how 30 participants were selected, how the data saturation was decided, add more details about how the thematic saturation was decided.

Line 115- add more details about how the thematic saturation was decided,

- Who conducted the interview?

- The interview was though with local language, what was the local language?

Linr 141- INVIVO-version?

Line 153- not clear... incomplete sentence..

Results:

- Present themes and sub-themes more systematically (e.g., thematic headings with clear structure).

- Under each theme, briefly summarize the key idea before providing quotes.

- Table 1 needs improved formatting.

- can add a summary table of themes.

Discussion:

- Add about what this study adds -a paragraph.

- Discuss potential systemic reasons behind poor communication (workload, staff shortages, power dynamics).

- Link results to respectful maternity care frameworks.

Strengths and limitations:

- Mention potential social desirability bias due to hospital setting.

- Clarify interviews were conducted by a non-clinician as a strength.

Conclusion:

- Write conclusions based on practical implications.

Others:

- Use Standardise terms: caesarean birth, caesarean delivery, and C-section.

- Over all Grammatical corrections:

Eg: This study novel provides insights… - This study provides novel insights…

- Summaries their demographic characteristics - summarises

- Need to edit small spacing/line break issues throughout the manuscript.

Note: in attachments, interview guide for medical practionners and codes are attached, but nothing related was mentioned in manuscript?

.

Reviewer #1: **Yes:** Rhona O'ConnellRhona O'ConnellRhona O'ConnellRhona O'Connell

Reviewer #2: No

---

## [Author Response · Author response to Decision Letter 1]

13 Feb 2026

Response to Reviewers

Manuscript ID: PONE-D-25-47624

Title: Women's experiences of communication with medical staff before and after emergency caesarean birth in Zambia: a qualitative study

Journal: PLOS ONE

Dear Editor and Reviewers,

We thank you and the reviewers for the constructive and detailed feedback provided. We have carefully addressed all comments and made the necessary revisions. Below, we provide point-by-point responses.

Response to Journal Requirements

1. PLOS ONE style requirements: Reformatted manuscript to comply with PLOS ONE style.

2. Inclusivity questionnaire: Completed and uploaded as Supporting Information.

3. Data availability contact: We have now included the anonymized transcripts as supporting information (see S5 and S7 Files) as well as revised to institutional contact: University of Zambia Biomedical Research Ethics Committee (unzarec@unza.zm). All personal identifiers have been removed to ensure participant confidentiality.

4. Abstract consistency: Ensured manuscript and online abstracts are identical.

5. Figure references: Added explicit references to Figure 1 and Figure 2 in the text.

6. Recommended citations: Added WHO (2018) and Bohren et al. (2015) on respectful maternity care.

7. Reference list: Reviewed; no retracted articles; added three new references.

Response to Reviewer #1

Comment 1 - Terminology consistency: Inconsistent use of caesarean birth/delivery.

Response: Standardised to "caesarean birth" throughout, retaining "delivery and caesarean section" only in direct quotes, referenced titles and anonymized transcripts.

Comment 2 - Table 1 formatting: Needs easier-to-read format.

Response: Completely reformatted Table 1 to present aggregated summary data by categories (age groups, parity, and education) with counts and percentages.

Comment 3 - Private healthcare: Did participants have the same level of care?

Response: Thank you for this helpful question. All participants were recruited from the same hospital and received care under the same clinical team and facility protocols. The reference standard (non-fee-paying) and premium wards relates only to where the interview was conducted, not to differences in clinical care received for the condition under study. We did not recruit participants based on ability to pay, insurance status, or use of private healthcare outside the hospital. We also did not collect or analyse private versus non-private healthcare status as a study variable, and the interviews did not aim to compare experiences by payment category. We used private rooms in both ward types to maximize confidentiality and reduce interruptions during interviews. To prevent misinterpretation, we have revised the sentence to clarify this point. Revised wording: To protect confidentiality, interviews were conducted in private treatment rooms or private rooms within the hospital wards, selected for privacy and minimal disruption, rather than to reflect differences in care or payment status. All participants received public healthcare at UTHWNH.

Comment 4 - "Starving" to "fasting": Changed "starving" to "fasting" as recommended.

Response to Reviewer #2

Abstract

Add design details: Added phenomenological qualitative design and inductive thematic analysis to abstract.

Specify key themes: Listed all four themes in abstract Results section.

Introduction

Global issues (violence, PTSD): Added paragraph on communication failures contributing to obstetric violence, mistrust, and PTSD.

Knowledge gap: Added explicit statement that no prior study examined this in Zambia's tertiary settings.

Materials and Methods

Sample size justification: Added justification citing qualitative guidelines (20-30 for phenomenological studies).

Thematic saturation: Explained iterative assessment; saturation reached at ~28 interviews; 2 additional to confirm.

Who conducted interviews: Clarified that the lead author conducted all interviews.

Local languages: Specified Bemba and Nyanja as the local languages.

NVivo version: Specified NVivo version 11.

Incomplete sentence (Line 153): Reviewed and corrected all incomplete sentences.

Results

Systematic theme presentation: Restructured with clear thematic headings, sub themes and brief summaries before quotes.

Table 1 formatting: Reformatted as a summary table with n and %

Summary table of themes: Added new Table 2 summarising themes and sub-themes.

Discussion

What this study adds: Added a paragraph on novel contribution.

Systemic reasons: Added section on workload, staff shortages, and power dynamics.

Respectful maternity care frameworks: Linked findings to WHO framework and RMC principles.

Strengths and Limitations

Social desirability bias: Added mention of potential bias due to hospital setting.

Non-clinician interviewer: Highlighted the lead author's non-clinician status as a strength.

Conclusion

Practical implications: Revised to include specific practical recommendations for training, language use, and policy.

Other Corrections

Standardised terms: Standardised to "caesarean birth" throughout.

Grammar ("novel provides"): Corrected to "provides novel insights."

Spelling ("summaries"): Corrected to "summarises."

Spacing/line breaks: Reviewed and corrected throughout.

Note on interview guide attachments: The interview guide for medical practitioners and codes were attached as supplementary materials for transparency. The manuscript focuses on women's experiences; discussions with healthcare providers were used to counter-check and contextualise the clinical experiences reported by patients as well as understand the hospital operations.

We trust these revisions adequately address the reviewers' concerns. We remain available for any further clarifications.

Sincerely,

Effie Moreblessing Mphande

On behalf of all co-author

---

## [Decision Letter · Decision Letter 1]

24 Mar 2026

Women’s experiences of communication with medical staff before and after emergency caesarean birth in Zambia: a qualitative study.

PONE-D-25-47624R1

Dear Dr. Mphande,

We’re pleased to inform you that your manuscript has been judged scientifically suitable for publication and will be formally accepted for publication once it meets all outstanding technical requirements.

Kind regards,

Sunita Panda, PhD

Academic Editor

PLOS One

Additional Editor Comments (optional):

Reviewers' comments:

Reviewer's Responses to Questions

**Comments to the Author**

Reviewer #1: All comments have been addressed

2. Is the manuscript technically sound, and do the data support the conclusions?

Reviewer #1: Yes

3. Has the statistical analysis been performed appropriately and rigorously?

Reviewer #1: N/A

4. Have the authors made all data underlying the findings in their manuscript fully available?

Reviewer #1: Yes

5. Is the manuscript presented in an intelligible fashion and written in standard English?

Reviewer #1: Yes

Reviewer #1: This manuscript is now acceptable for publication, all suggested edits have been addressed. I hope that communication for health care professionals will improve once the findings from this study are disseminated.

.

Reviewer #1: No

---

## [Editor Report · Acceptance letter]

PONE-D-25-47624R1

PLOS One

Dear Dr. Mphande,

I'm pleased to inform you that your manuscript has been deemed suitable for publication in PLOS One. Congratulations! Your manuscript is now being handed over to our production team.

Kind regards,

on behalf of

Dr Sunita Panda

Academic Editor

PLOS One